# Functionalized Graphene from Electrochemical Exfoliation of Graphite toward Improving Lubrication Function of Base Oil

Chunfeng Zhang [1], Xiaojun Zhang [1], Wei Zhang [1], Zhuang Zhao [2] and Xiaoqiang Fan [2,*]

[1]  Taihang Lubricant Technology Co., Ltd., Changzhi 046100, China
[2]  Key Laboratory of Advanced Technologies of Materials (Ministry of Education), School of Materials Science and Engineering, Southwest Jiaotong University, Chengdu 610031, China
[*]  Correspondence: fxq@home.swjtu.edu.cn

**Abstract:** Electrochemical exfoliation of graphene is an environmentally friendly method, which enables mass production. Herein, three ionic liquids (ILs) with the same imidazole cation were used to exfoliate graphite into functionalized graphene, as a lubricant additive in an acetonitrile solution. Chemical and structural characterization revealed the relationship between the functionalization density of graphene and the concentration of IL, showing higher concentrations with higher densities. The exfoliated graphene hybrid oil displayed good dispersion because of a high functionalization density. More importantly, the different anions affected the tribological properties of the exfoliated graphene. Among them, the exfoliated graphene with 1-butyl-3-methylimidazolium hexafluorophosphate ([BMIm][PF$_6$]) possessed the best tribological performance, and the average friction coefficient and wear volume were reduced by 32% and 39%, respectively. Through the characterization of worn surfaces and wear debris, the lubrication mechanism and structural evolution of the functionalized graphene were illustrated in detail. The good lubrication function was attributed to the formation of a tribo-film and the disorder transformation of the graphene structure. The simultaneous exfoliation and functionalization of graphene offers a promising advanced lubricant for industrial fabrication.

**Keywords:** electrochemical exfoliation; graphene; ionic liquids; lubricant additives; tribological properties

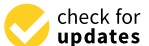



## 1. Introduction

Graphene is a two-dimensional carbon nanomaterial, consisting of carbon atoms in a hexagonal honeycomb lattice with sp$^2$ hybridized orbitals and considered to be a revolutionary material of the future [1]. The literature on graphene has reported groundbreaking results in many fields, including biology [2], energy [3], and catalysis [4]. Among these, graphene as a lubricating material in tribology has attracted much attention from academia and industry for its unique structure and performance advantages [5–7]. The interlayer weak shearing and high mechanical strength of graphene allows it to meet the characteristic requirements of a lubricating additive [8,9]. In addition, the thermal conductivity of graphene is beneficial to the emission of the heat generated by friction [10]. Therefore, graphene as a lubricating additive has a significant potential in friction reduction and anti-wear applications. However, despite the above advantages, the high preparation cost and poor dispersion stability of graphene are shortcomings for industrial application compared with the traditional lubricating additives [11].

Since the exfoliation of graphene in 2004 [12], many graphene preparation methods have been devised. For "bottom-up" preparation methods, epitaxial growth [13] and chemical vapor deposition [14] can produce high-quality graphene, but their high cost and low yield make it difficult to achieve industrial application. The "top-down" preparation methods, such as mechanical exfoliation [12,15], redox [16–18], chemical exfoliation [19–22], and liquid phase exfoliation [23,24], are easier to implement at a large scale and have a

lower cost, attracting more attention for practical applications. Among them, the chemical redox method is relatively simple, and the monolayer rate of graphene is high, which ensures a wide range of applications. At present, chemical redox is the most widely used method for the preparation of graphene at this stage. However, many experimental results have demonstrated that redox preparation methods require the use of toxic reagents, as well as being time-consuming. Furthermore, strong acids and strong oxidants have to be used in the preparation process, which is not only disadvantageous to the environment but also seriously damages the lattice structure of graphene, which even after reduction cannot be completely repaired. Therefore, finding a relatively green and efficient preparation method for graphene remains an important research direction for large-scale preparation of graphene.

Compared with the traditional exfoliation method, the electrochemical exfoliation method has the advantages of low cost, room temperature operation, in situ functionalization, and so on [25]. In addition, the electrochemical exfoliation method can optimize preparation parameters by adjusting the concentration and voltage of the electrolyte [26]. Using specific electrolytes, anions and cations can be introduced into the graphene structure, and functional graphene can be formed while the graphene is stripped [27]. Different anion and cationic structures give functionalized graphene a high dispersion stability in polar or non-polar solutions [28]. For the internal electrochemical preparation of graphene in organic electrolyte, exfoliation products may be formed on both the anode and cathode, which is related to the selected electrolyte solvent and solute and other influencing factors. Moreover, for electrolyte solvents, organic reagents with a large electrochemical window, such as acetonitrile, N-methyl pyrrolidone (NMP), and dimethyl sulfoxide (DMSO) are generally chosen. In the process of exfoliation, the anions and cations in the electrolyte are directly embedded in the graphite layer under the action of the voltage, and the graphite raw materials are expanded, which can also realize the functionalization of graphene; and then, the graphene is dispersed using ultrasound or other means [29]. During the exfoliation process, the possibility of graphene products becoming oxidized is very small, which is also a main advantage of electrochemical exfoliation of organic electrolytes in ensuring the quality of modified graphene products [30]. Meanwhile, the low degree of oxidation also greatly improves the application range of modified graphene as an additive in the field of lubrication [31].

Herein, acetonitrile solution was selected as an electrolyte solvent. Graphene with the same imidazole cation was prepared using three ionic liquids (ILs), including 1-butyl-3-methylimidazolium hexafluorophosphate ([BMIm][PF$_6$]), 1-Butyl-3-methylimidazolium tetrafluoroborate ([BMIm][BF$_4$]), and 1-Butyl-3-MethylImidazolium bis (trifluoromethyl-sulfonyl) imide ([BMIm][NTf$_2$]), as electrolyte solutes. These three ILs have long been used in the study of modified graphene additives, but no scholars have stripped graphene through the electrochemical exfoliation of graphene in an organic solution system for use as a lubricating additive. Therefore, we used them as preferred ILs for electrochemical exfoliation of graphene in organic solutions. In this work, graphene exfoliated with three kinds of ILs with different concentrations was obtained, and the differences in morphology, structure, and composition among the samples were compared using a series of characterization methods. Then, the exfoliated graphene was added to 150 N base oil as an additive, and their different tribological performances were studied and analyzed. Through the characterization of wear marks and debris, the lubrication mechanism and evolution law of exfoliated graphene at the contact interface are explained in detail.

## 2. Materials and Methods

### 2.1. Materials

High-purity graphite rod (99.99%) was purchased from Beijing Synzong Tianqi New Material Technology Co., Ltd. (Beijing, China) [BMIm][PF$_6$] (99%), [BMIm][BF$_4$] (99%), and [BMIm][NTf$_2$] (99%) were obtained from Lanzhou Institute of Chemical Technology, Chinese Academy of Sciences (Lanzhou, China). The 150N base oil was purchased from

Liugong Machinery Co., Ltd. (Guangxi, China). Petroleum ether, anhydrous ethanol, and NMP were supplied by Chengdu Kelong Chemical Co., Ltd. (Chengdu, China). All chemical reagents were analytical grade and used directly.

### 2.2. Electrochemical Exfoliation of Graphene

Before beginning the experiments, the graphite rod was cleaned with deionized water, to remove the large particles on the surface. The steps of the electrochemical exfoliation process were as follows:

(a)   Three ILs, [BMIm][PF$_6$], [BMIm][BF$_4$], and [BMIm][NTf$_2$], were mixed with acetonitrile solvents. Electrolytes of different concentrations were prepared with volume ratios of IL and acetonitrile solvents at 1:10, 1:30, and 1:50, respectively. The molecular structures of the three ILs s are shown in Figure 1. Then, electrochemical exfoliation was carried out in the traditional two-electrode system, using graphite rods as cathodes and anodes, keeping the two electrodes parallel and 4 cm apart from each other, and connecting a direct-current (DC) power supply.

(b)   A DC voltage of 15 V was applied at both ends of the electrode. The electrolyte was colorless and transparent at first, but after 30 min, the anode graphite gradually decomposed and peeled off into the electrolyte, the color of the solution began to change, the particles separated from the graphite matrix appeared at the bottom, and the solution blackened completely after 2 h of exfoliation.

(c)   In 50 mL electrolyte, NMP of 50 mL was added to the exfoliated suspension and a voltage of 15 V was applied at both ends for 2 h. Then the supernatant was centrifuged at 12,000 rpm, after ultrasound for 1 h. The collected products were washed with anhydrous ethanol 3 times and then dried in a vacuum drying box at 60 °C for 36 h. The products are shown in Table 1.

**Table 1.** Names of the electrochemical stripping products from the ILs and acetonitrile solvent organic systems.

| ILs | 1:10 | 1:30 | 1:50 |
|---|---|---|---|
| [BMIm][PF$_6$] | P$_{10}$ | P$_{30}$ | P$_{50}$ |
| [BMIm][BF$_4$] | B$_{10}$ | B$_{30}$ | B$_{50}$ |
| [BMIm][NTf$_2$] | N$_{10}$ | N$_{30}$ | N$_{50}$ |

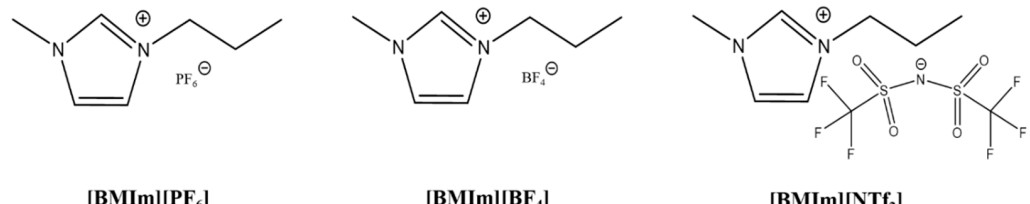

**Figure 1.** Molecular structures of [BMIm][PF$_6$], [BMIm][BF$_4$], and [BMIm][NTf$_2$].

### 2.3. Tribological Characterization

The friction-reducing and anti-wear properties of 150 N lubricating oil with additive were tested in a friction test. The nine kinds of graphene product were added to 150 N base oil at 0.08 wt.%. After ultrasonic treatment for 2 h, the friction experiments were carried out using a self-built sliding friction tester in our laboratory. The diameter of GCr15 bearing steel as the upper steel ball (HRC 62) was 10 mm. The fixed lower steel plate (HRC 63, Ra = 28 nm) consisted of AISI 52,100 steel ($\Phi$24 × 7.9 mm). At room temperature, the load of 60 N (Hertz contact pressure 1147 MPa) was applied at a frequency of 5 Hz, and the displacement amplitude was set to 1 mm. The profile and wear volume of the wear tracks were detected with a white light interferometer (WLI, Contour GT, Germany), and the morphology of the worn surface was observed using a scanning microscope

(SEM, JSM-6610LV). Then the worn surfaces lubricated by hybrid oils were analyzed by Raman spectroscopy, and the wear debris under lubricated by $P_{10}$ was collected, to observe the morphology under field emission SEM (FESEM), to further explore the lubrication mechanism of the graphene additive. In addition, the wear rate (*WR*) was calculated using the following equation:

$$WR = \frac{V}{F \cdot L}$$

where *V* is the wear volumes, *F* is the applied load, and *L* is the whole sliding distance.

## 3. Results and Discussion

### 3.1. Chemical and Structural Characterization of the Products

In order to observe the morphology of graphene obtained by electrochemical exfoliation of the organic solution, all samples were characterized using FESEM (Figure 2). In the FESEM images at the same ratio, it can be seen that the graphene samples prepared through organic solution exfoliation had no significant difference in size. The graphene samples showed a flat lamellar shape, indicating the successful exfoliation of graphene with a lower degree of surface defects, complete preservation of plane structure, and a lower degree of oxidation. In addition, through the observation of the FESEM images, multilayer stacked structures can clearly be seen at the edges of these graphene samples, indicating that the prepared products may be multilayer graphene nanosheets.

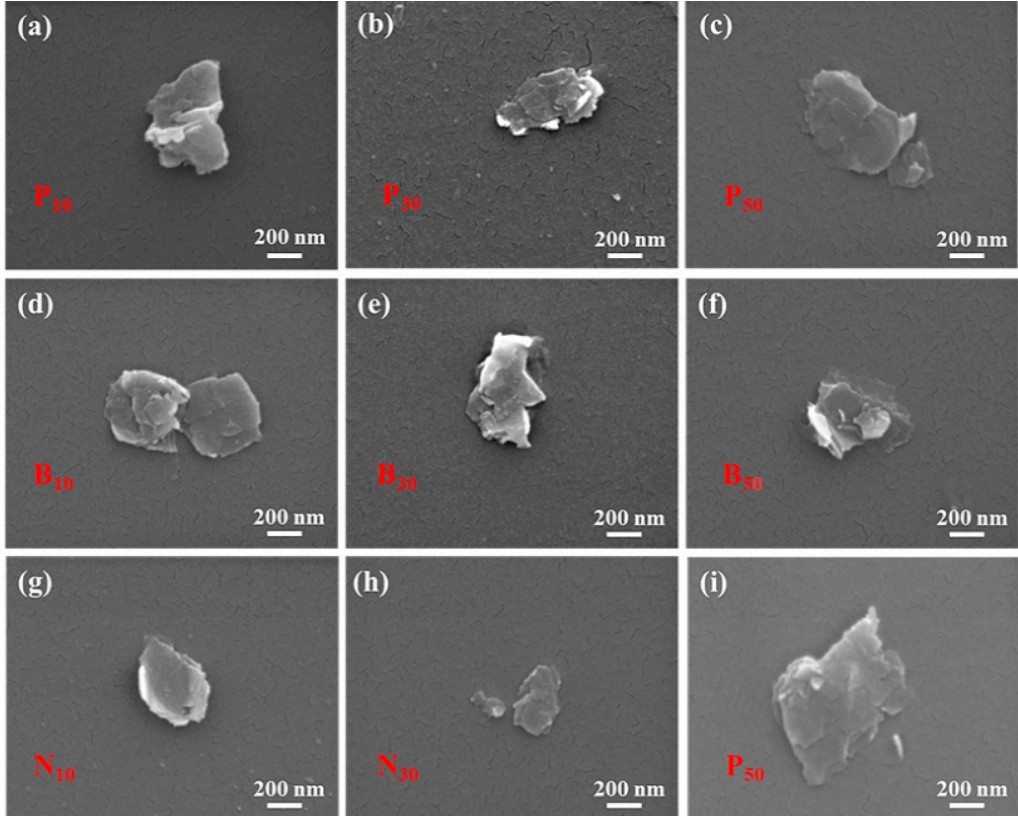

**Figure 2.** FESEM diagrams of graphene exfoliated by organic solutions with (**a**–**c**) [BMIm][PF$_6$], (**d**–**f**) [BMIm][BF$_4$], and (**g**–**i**) [BMIm][NTf$_2$] at 1:10, 1:30, and 1:40, respectively.

The X-ray photoelectron spectroscopy (XPS) spectra of the products are illustrated in Figure 3. The elemental composition of the exfoliated graphene can be clearly seen in the spectra. All samples contained the elements of C, N, O, and F, but the contents of each element were different. Table 2 lists the typical elements and their contents determined by XPS. Combined with XPS full spectrum analysis, the nitrogen and fluorine contents of

the modified graphene exfoliated using [BMIm][PF$_6$], [BMIm][BF$_4$], and [BMIm][NTf$_2$] increased with the increase of the concentration of ILs organic solution, indicating that the higher the concentration of organic solution, the higher the density of graphene modification. In addition, compared with the initial graphite, the oxygen content of exfoliated graphene in [BMIm][PF$_6$] and [BMIm][BF$_4$] IL organic solution decreased, and the higher the electrolyte concentration, the lower the degree of oxidation. This mainly depended on the grafting and adsorption of the modified ions, which increased the number of surface and internal groups of graphene. Graphene cannot be further oxidized, which greatly limits the degree of oxidation of graphene [32]. The higher the concentration, the stronger the grafting and adsorption of the modified ions, thus producing a lower oxygen content. On the contrary, with the increase of [BMIm][NTf$_2$] electrolyte concentration, the oxygen content and oxidation degree of graphene after exfoliation increased obviously. This was mainly caused by the oxygen functional groups in the [BMIm][NTf$_2$] IL. In summary, the oxygen contents of the exfoliated graphene from [BMIm][PF$_6$] and [BMIm][BF$_4$] ILs were lower, and the oxygen content was inversely proportional to the electrolyte concentration, while the oxygen content of the [BMIm][NTf$_2$] IL was opposite, because of the oxygen groups. Fine spectra of the N1s and F1s of exfoliated graphene using the three kinds of IL with different concentrations are shown in Figure 4. The intensity of the peaks in the fine spectrum of nitrogen and fluorine increased with the increase of the concentration of organic solution, which further verified that the higher the concentration of the organic solution, the higher the density of graphene modification.

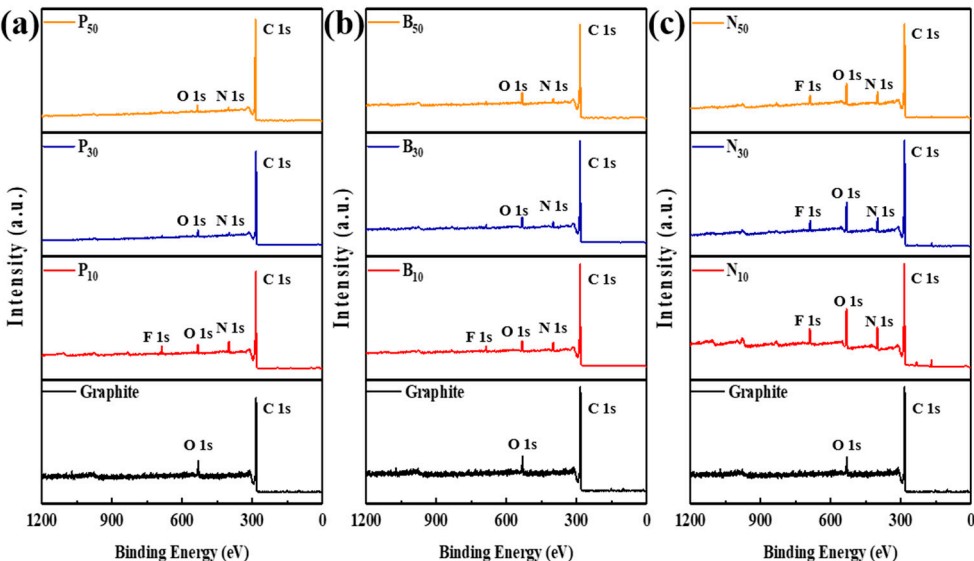

**Figure 3.** XPS full spectra of initial graphite and exfoliated graphene in organic solution of (**a**) [BMIm][PF$_6$], (**b**) [BMIm][BF$_4$], and (**c**) [BMIm][NTf$_2$] at 1:10, 1:30, and 1:50, respectively.

The phase and grain size of the products were further explored through XRD analysis, as shown in Figure 5. The state and interlayer situation of the graphene exfoliated with organic solution at different concentrations were compared and analyzed. The diagram shows that the initial graphite has an obvious peak at $2\theta = 26.64°$, corresponding to the (002) crystal plane, with an inner layer spacing of 0.334 nm. Compared with the (002) crystal plane peak of the initial graphite, the peak intensity of all the exfoliated samples decreased by two orders of magnitude, but the peak width remained unchanged, which indicated that the lamellar structure of long-range stacking of graphite collapsed in the electrochemical process [33]. In other words, in the organic solution of the ILs, the modified graphene was successfully exfoliated on the graphite matrix by electrochemical means, and the exfoliated graphene nanosheets had relatively complete crystal structures.

**Table 2.** Typical components and contents of graphene stripped from the initial graphite and organic solution.

| Samples | C% | N% | O% | F% |
|---|---|---|---|---|
| graphite | 92.34 | — | 7.66 | — |
| $P_{10}$ | 84.28 | 8.23 | 3.25 | 3.45 |
| $P_{30}$ | 85.91 | 7.65 | 3.49 | 2.38 |
| $P_{50}$ | 87.61 | 7.02 | 3.72 | 1.36 |
| $B_{10}$ | 85.31 | 7.18 | 4.06 | 2.58 |
| $B_{30}$ | 86.5 | 6.33 | 4.24 | 1.99 |
| $B_{50}$ | 88.38 | 5.41 | 4.55 | 1.29 |
| $N_{10}$ | 74.42 | 7.42 | 11.12 | 4.24 |
| $N_{30}$ | 76.14 | 7.05 | 10.46 | 3.35 |
| $N_{50}$ | 78.16 | 6.57 | 9.58 | 2.19 |

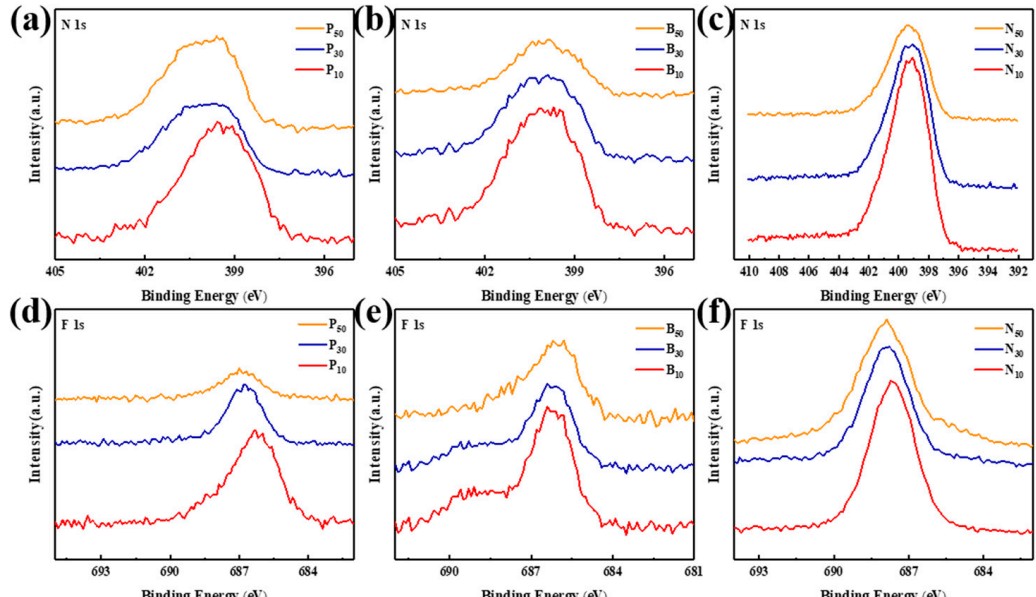

**Figure 4.** XPS fine spectra on typical N and F elements from exfoliated graphene in organic solution of (**a**,**d**) [BMIm][PF$_6$], (**b**,**e**) [BMIm][BF4], and (**c**,**f**) [BMIm][NTf$_2$], respectively.

The magnification curves of graphene (002) crystal plane exfoliated using organic solutions at different concentrations were analyzed. The position of the diffraction peak of the (002) crystal plane of all graphene samples tended to shift to the left relative to the initial graphite. The results display that the distance between the graphite crystal planes increased during the process of electrochemical exfoliation. The reason for these results may have been the intercalation and grafting of ions. Moreover, some of the embedded ions may acted as a modifier of the graphene. The higher the concentration of IL, the stronger the modification effect, so the diffraction peak of the exfoliated graphene (002) crystal plane shifted to the left. Although the graphene electrochemically modified using ILs retained the original graphite lamellar structure, the intercalation and modification of ions increased the interlayer spacing of graphene. In a certain range, the higher the concentration of IL, the more the interlayer spacing increased.

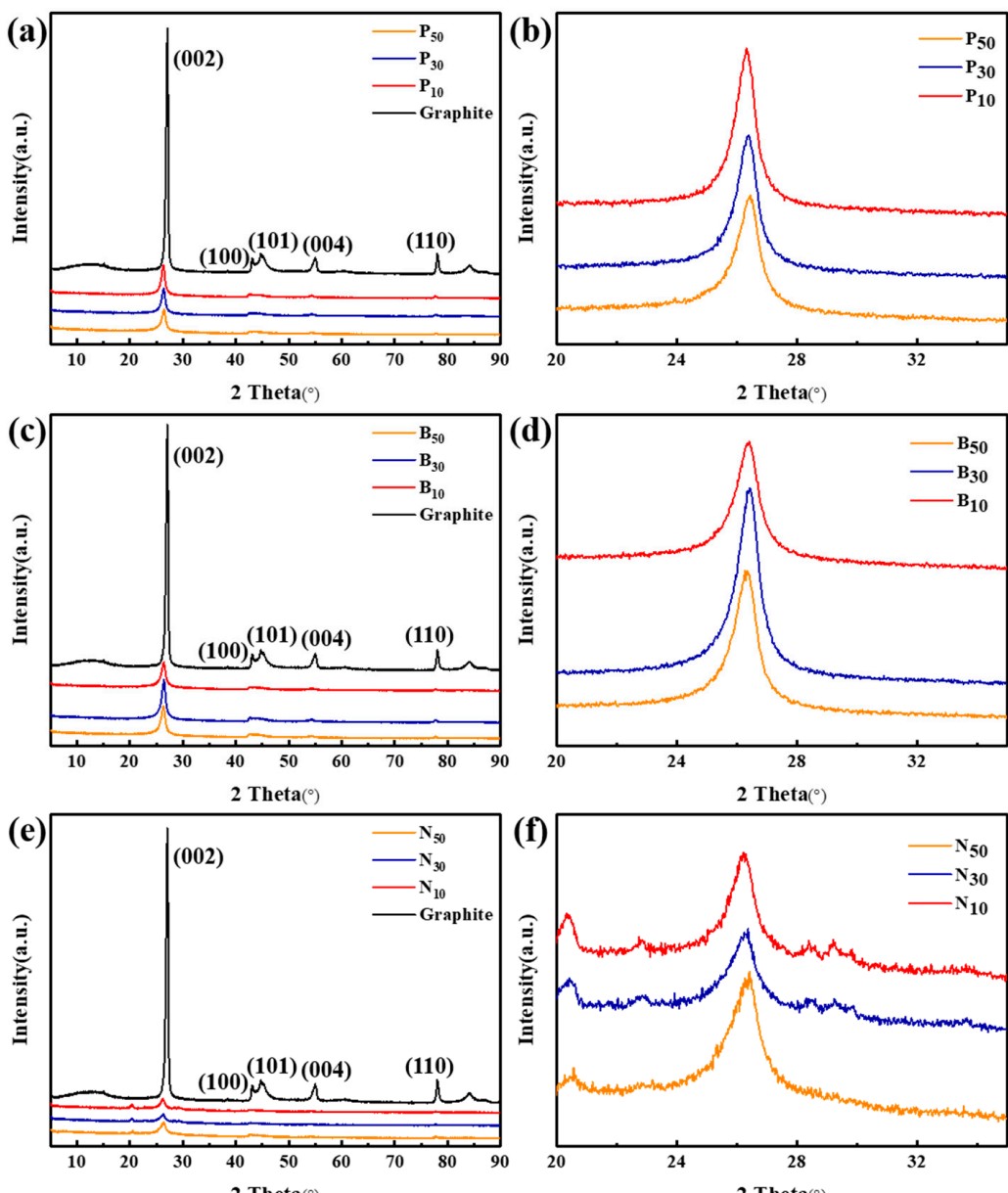

**Figure 5.** (**a,c,e**) XRD curves and (**b,d,f**) magnification curves (002) crystal plane) of initial graphite and graphene exfoliated using organic solution of [BMIm][PF$_6$], [BMIm][BF$_4$], and [BMIm][NTf$_2$] at 1:10, 1:30, and 1:50, respectively.

Raman spectra analysis was performed with a Raman microscope, as seen in Figure 6. In general, the graphite curves show three typical characteristic peaks, namely, the D peak at 1350 cm$^{-1}$, the G peak at 1570 cm$^{-1}$, and the 2D peak at 2700 cm$^{-1}$ [34]. Quantitative analysis of the degree of defects in the samples was realized by calculating the intensity ratio of D peak to G peak ($I_D/I_G$) [35], which can be related to the density of in situ modification. As shown in Figure 6, the $I_D/I_G$ values of the graphene exfoliated with the three kinds of IL all increased with the increase of IL concentration, which indicates that the samples exfoliated with a high concentration had a higher modification density, resulting in a higher degree of defects.

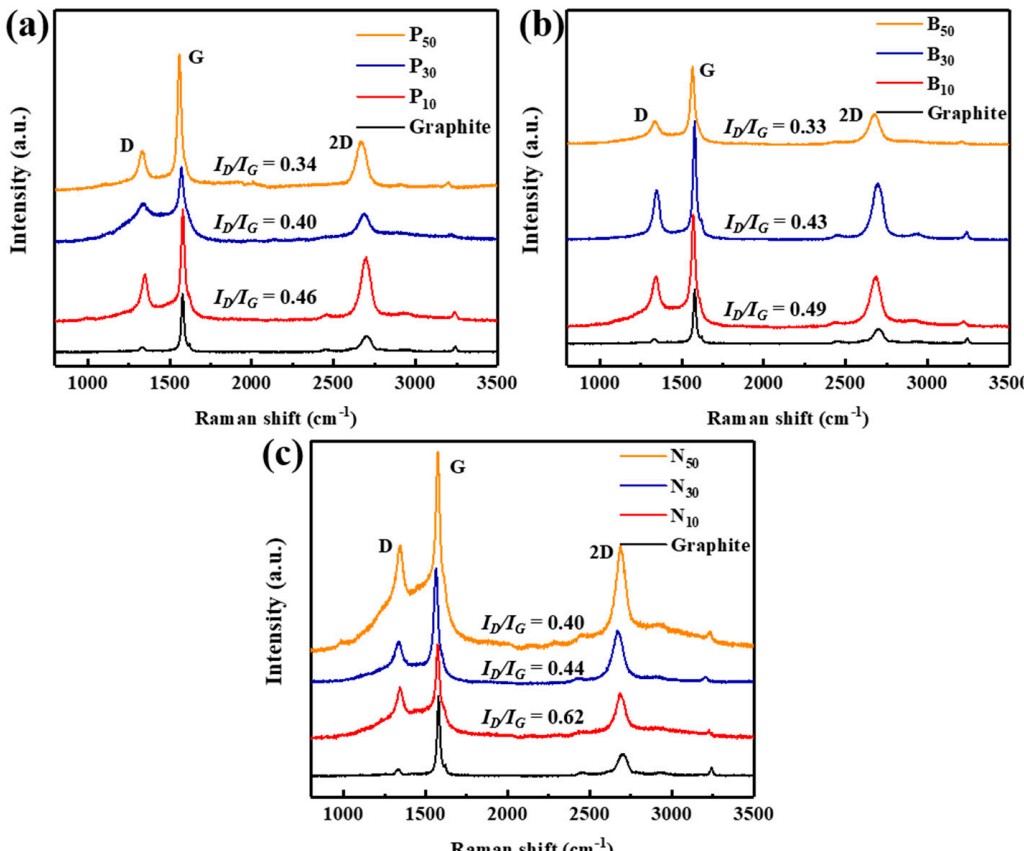

**Figure 6.** Raman curves of the initial graphite and graphene exfoliated using organic solutions of (**a**) [BMIm][$PF_6$], (**b**) [BMIm][$BF_4$], and (**c**) [BMIm][$NTf_2$] at 1:10, 1:30, and 1:50, respectively.

In order to further explore the modified groups on the surface of the graphene stripped using an organic solution, the samples were analyzed with a Fourier transform infrared spectrometer (FTIR), as shown in Figure 7. The peak was the stretching vibration peak of the C–H bond corresponding to 2875 and 2935 $cm^{-1}$ (including the symmetrical methylene and asymmetric methyl on the alkyl chain in the ILs), which provided a strong basis for the modification of graphene using the ILs [36]. At the same time, compared with the initial graphite, it was found that the shear vibration of the H–C=C–H group on the imidazole ring in the ILs and the strength of the product at all high IL concentrations was higher, which indicates that the density of the imidazole ring group on the surface may have been higher. Based on the above analysis, it can be determined that IL functional groups existed between the layers of the multi-layer graphene, and the graphene exfoliated ($P_{10}$, $B_{10}$ and $N_{10}$) at high IL concentrations had a lower oxidation rate and higher ion modification density.

Thermogravimetric analysis (TGA) was used to investigate the thermal stability of the exfoliated graphene, as shown in Figure 8. Under the condition of $N_2$ flow protection, a sample was heated from room temperature to 1000 °C, with the heating rate of 10 °C/min. Separation of the adsorbed water occurred in the first stage (30 °C–120 °C), which mainly came from the water in the air. In the second stage, mainly $CO_2$ and CO were released, which resulted from the elimination of oxygen functional groups on the surface of the graphene and the pyrolysis of the carbon skeleton [37,38]. It can be seen that $N_{10}$, $N_{30}$, and $N_{50}$ all had a high oxygen content. The mass loss in the final stage was related to the thermal decomposition of the modified groups on the graphene samples. The mass loss in this stage indicated that the modified groups in the ILs were connected to the modified graphene surface or embedded in the modified graphene layer in a certain form of interaction (chemical reaction and physical adsorption), which effectively modified the

graphene products. The mass loss of the $P_{10}$, $B_{10}$, and $N_{10}$ samples in the third stage was greater than that of the low concentration products of the same IL, indicating that the exfoliated products had more modified groups at high concentration.

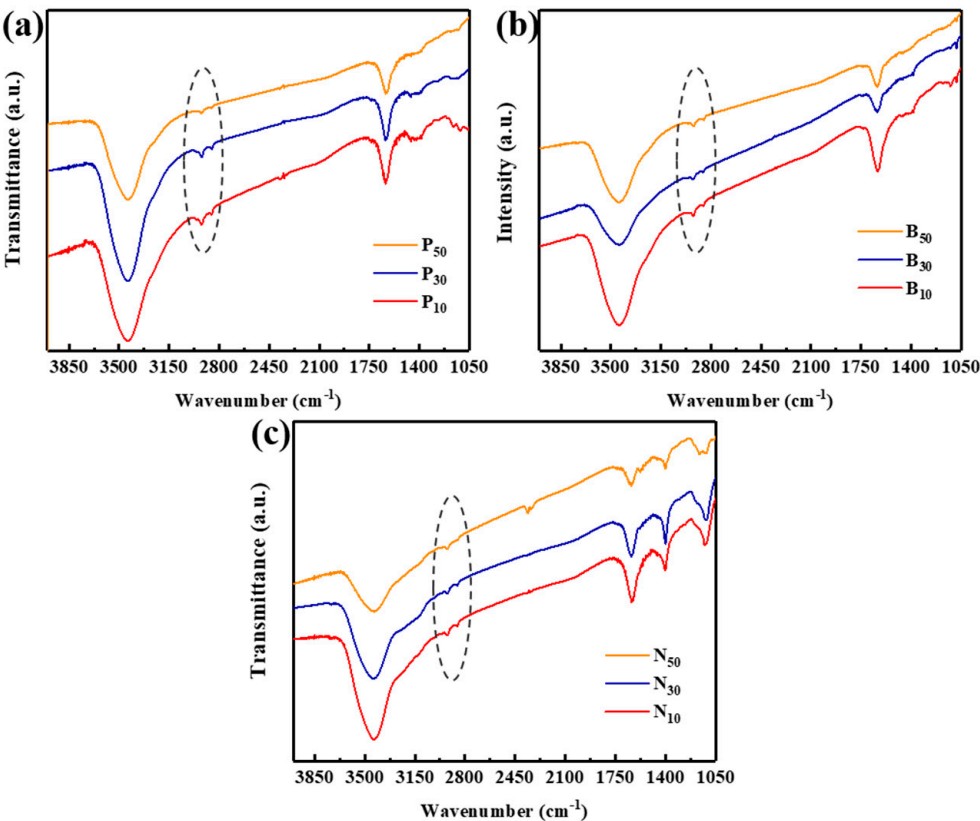

**Figure 7.** FTIR curves of the initial graphite and graphene exfoliated using organic solutions of (**a**) [BMIm][PF$_6$], (**b**) [BMIm][BF$_4$], and (**c**) [BMIm][NTf$_2$] at 1:10, 1:30, and 1:50, respectively.

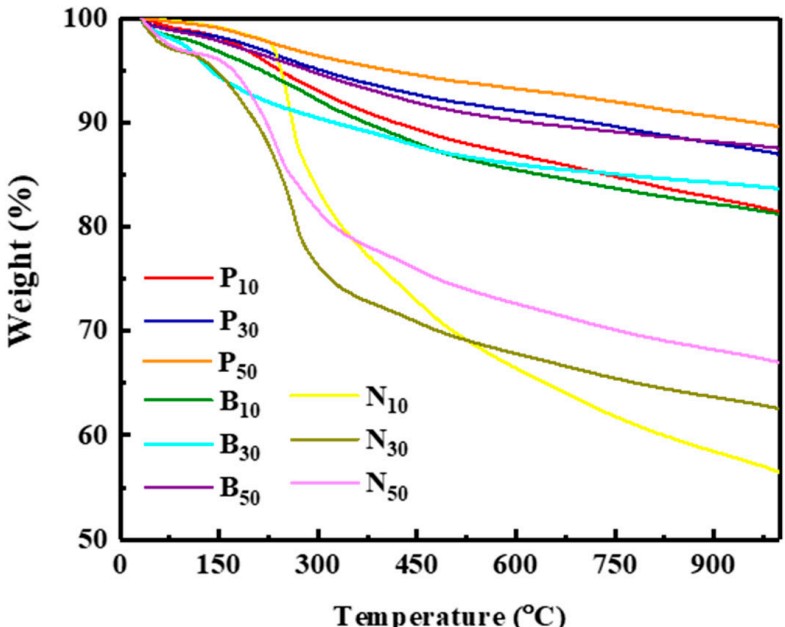

**Figure 8.** TGA curves of graphene exfoliated using organic solutions with [BMIm][PF$_6$], [BMIm][BF$_4$], and [BMIm][NTf$_2$] at 1:10, 1:30, and 1:50, respectively.

### 3.2. Mechanism Analysis of the Electrochemical Exfoliation of Graphene from IL Organic Solutions

To better understand the process of the electrochemical exfoliation of graphene with organic electrolytes, the mechanisms of the electrochemical exfoliation of graphene in an aqueous solution of IL are illustrated in Figure 9. Instead of using water as an electrolytic solvent, the oxidation of the modified graphene was greatly reduced with the use of an organic solvent, and the process was mainly divided into three steps.

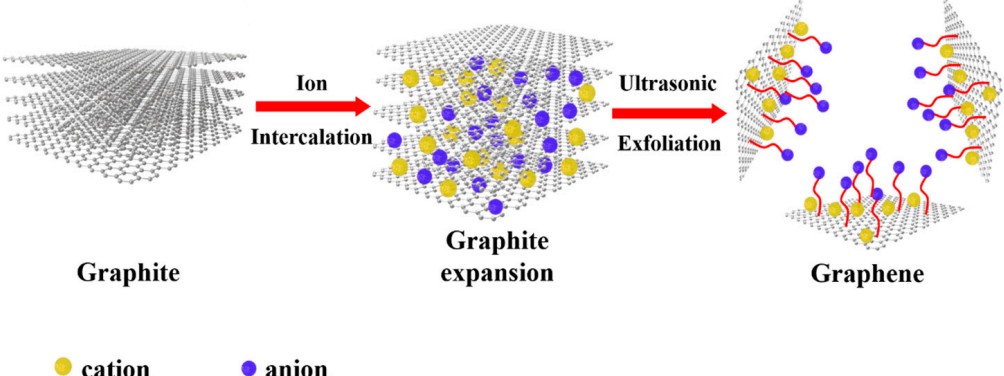

**Figure 9.** Mechanism diagram of the electrochemical stripping of graphene from the IL organic solution.

First, under the action of the 15 V voltage, the anions of ILs began to approach the anode and became embedded in the graphite interlayer, and the electrochemical oxidation reaction occurred and covalently connected to the graphite matrix. On the other hand, the cations around the anode became attached to the graphite surface and interlayer through a π-π conjugation on the graphene surface. This process also produced defects in the graphene layer. Then, the anodic graphite expanded, and the interaction between the ions of the ILs and the graphite surface changed the structure and composition of the graphite surface; that is, the graphite surface was grafted or attached to many groups. With the accumulation of graphite interlayer groups, the interlayer was gradually opened, resulting in the expansion of the graphite rods to a certain extent and formation of cracks, and over time, the cracks gradually increased. Finally, sufficient power was provided for the exfoliation of the modified graphene. In this work, the interlayer van der Waals force was overcome and the graphene was diffused into the solution through an ultrasound-assisted graphene lamellar process.

### 3.3. Mechanism Analysis of the Electrochemical Exfoliation of Graphene from the IL Organic Solutions

The exfoliated graphene powder was added to the base oil and dispersed using ultrasonication. The exfoliated graphene hybrid oils were left for two weeks at room temperature to verify their dispersion stability, as shown in Figure 10. It can be seen that the hybrid oils containing modified graphene exfoliated using [BMIm][PF$_6$] and [BMIm][BF$_4$] ILs showed good dispersion. On the other hand, the exfoliation products of [BMIm][NTf$_2$] IL were delaminated in the base oil, which may have been due to the thicker lamella and larger size of the exfoliated products.

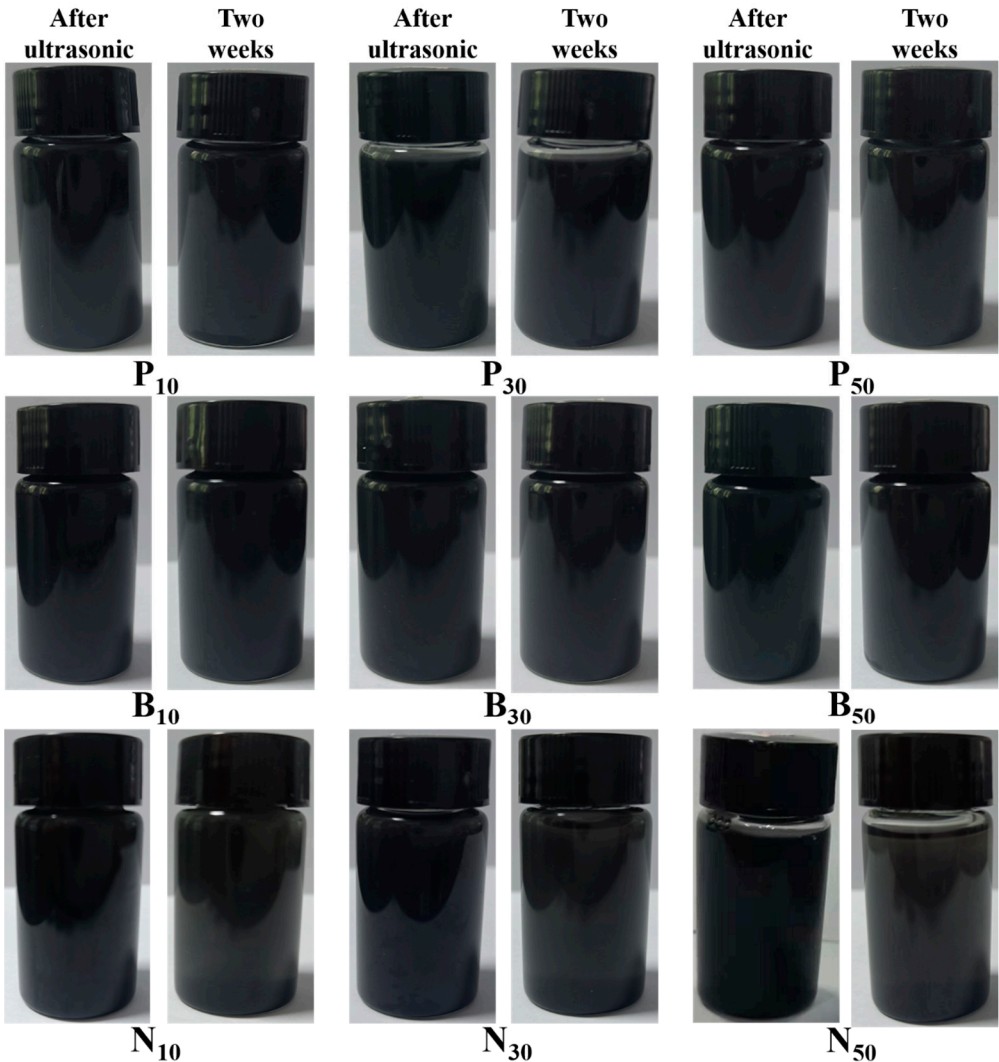

**Figure 10.** Dispersion of the exfoliated graphene hybrid oils, after standing for two weeks.

### 3.4. Tribological Performance of the Exfoliated Graphene

The tribological properties of the graphene exfoliated from organic solutions, as additives to base oil, were tested using a friction test. Friction experiments of ball–disk contact at an applied load of 60 N were carried out for all lubricating oil samples for 1 h at room temperature. Friction coefficient curves with time are shown in Figure 11. In the experiment of 150 N base oil, due to the failure of the oil film, the friction coefficient increased from the beginning, then fluctuated, and finally stabilized at around 0.15. The friction coefficient of the exfoliated graphene hybrid oils decreased obviously, because the modified graphene additives in the lubricating oil entered into the friction interface, to form a protective film. From the friction curves, it can be seen that the friction-reduction performance was greatly affected by the type of IL, but the effect of the concentration of exfoliate was not obvious. Specifically, the graphene exfoliated with [BMIm][PF$_6$] IL had a lower friction coefficient (around 0.10), meaning that the friction-reduction performance of P$_{10}$ was the best. The reason for the above results may have been that P$_{10}$ had a thinner layer, which could more easily enter into the friction interface, to form a friction transfer film, thus effectively reducing the friction coefficient. On the contrary, the friction-reducing properties of graphene modified using [BMIm][NTf$_2$] IL at different concentrations were not ideal, and the friction coefficient fluctuated, which may have also been affected by the thickness of the lamellas.

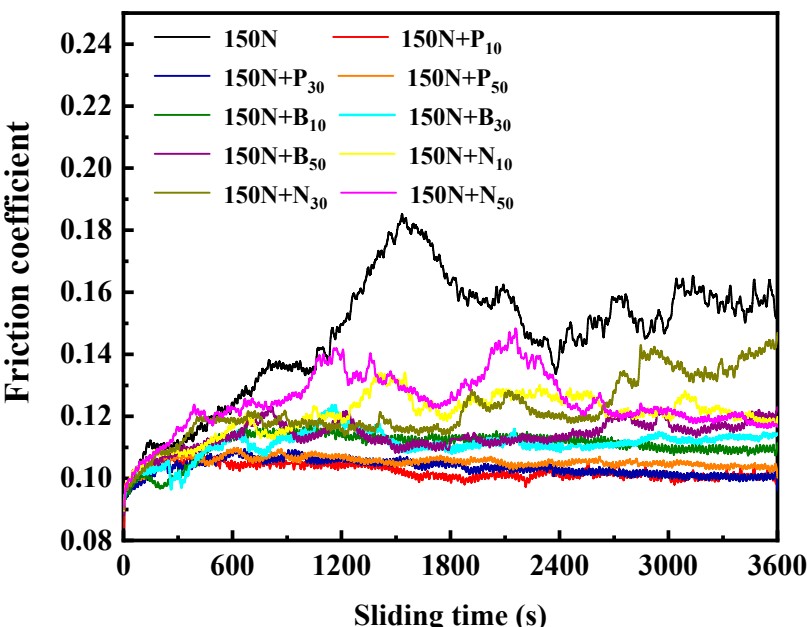

**Figure 11.** Friction coefficient curves of the base oil and exfoliated graphene hybrid oils.

The *WR* of the worn surface was measured using WLI. Figure 12 shows the average friction coefficient (AFC) and *WR* of each sample after the friction experiment, which can further demonstrate the wear resistance of the graphene exfoliated using IL organic solutions. Compared with the base oil, the AFC and WR of the exfoliated graphene hybrid oils were greatly reduced, showing a certain wear resistance. At the same time, similarly to the friction coefficient, the *WR* was also greatly affected by the type of IL, and the graphene exfoliated with [BMIm][$PF_6$] IL had a better wear resistance. Compared with the base oil, the $P_{10}$ hybrid oils reduced the AFC and *WR* by 32% and 39%, respectively.

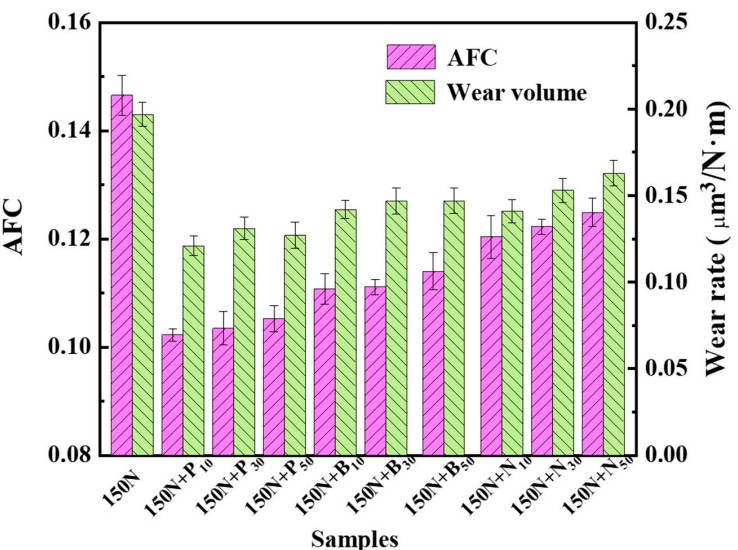

**Figure 12.** AFC and *WR* of the base oil and exfoliated graphene hybrid oils.

To investigate the applicability of the additive in actual working conditions, friction experiments for the $P_{10}$ hybrid oil were carried out at 40 °C and 100 °C. The friction curves for AFC and *WR* are shown in Figure 13. As shown in Figure 13a, the friction coefficient of the $P_{10}$ hybrid oil was lower than that of the base oil at 40 °C and 100 °C, which effectively enhanced the lubrication performance of the base oil. In addition, Figure 13b shows the AFC and *WR* at high temperature. It can be seen that the AFC of the $P_{10}$ hybrid oil decreased

by 17.4% and 14.6% at 40 °C and 100 °C, respectively, and the *WR* decreases by 39.5% and 44.3%, respectively. These results show that $P_{10}$ had an obvious friction-reduction and wear resistance at higher temperature and showed good applicability.

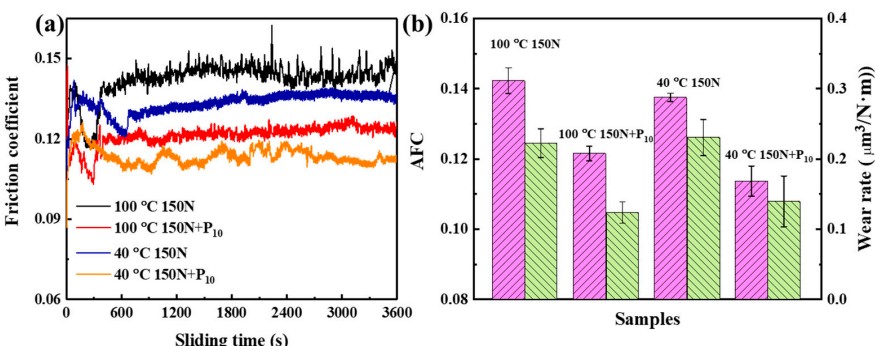

**Figure 13.** (**a**) Friction curves, (**b**) AFC and *WR* of the base oil and $P_{10}$ hybrid oil.

Subsequently, the morphologies of the worn surface after the friction test were observed using SEM, as shown in Figure 14. It can be seen in Figure 14j that the worn surface of the 150 N base oil had deeper parallel furrows and more flake debris at the center of the wear track. After adding the different exfoliated graphene additives, the morphologies of the wear tracks improved to varying degrees. The number of furrows was obviously reduced and the depth was shallower. Therefore, compared with the 150 N base oil, the addition of exfoliated graphene effectively improved the wear resistance of the base oil. The graphene exfoliated using the [BMIm][PF6] IL showed excellent wear resistance.

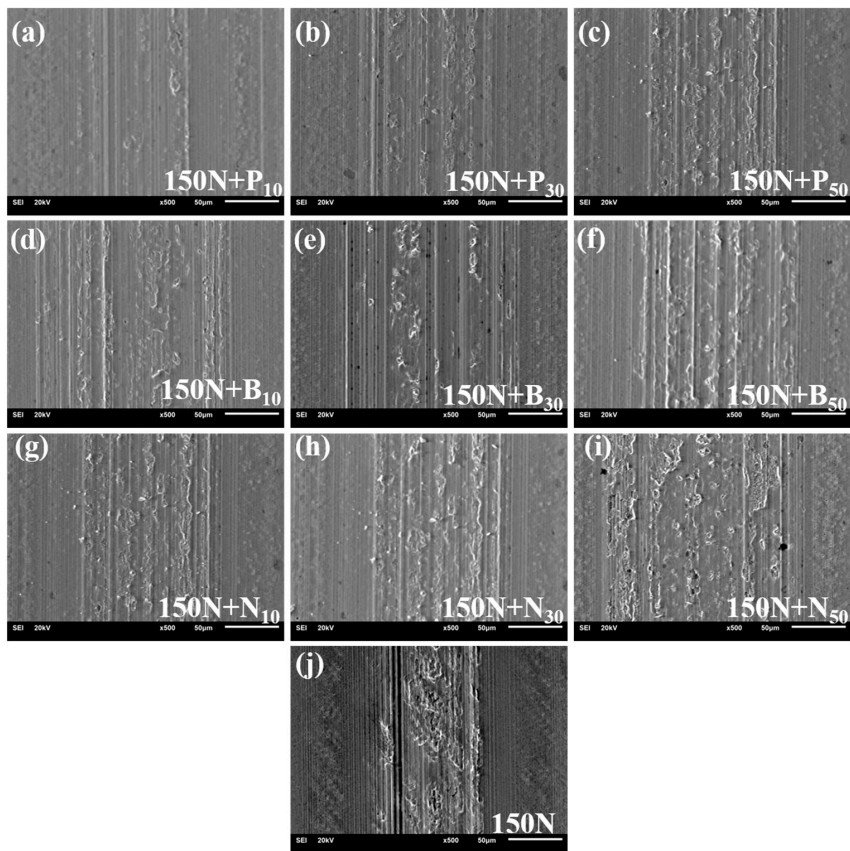

**Figure 14.** SEM images of the morphologies of the worn surfaces lubricated using (**a**–**c**) [BMIm][PF$_6$], (**d**–**f**) [BMIm][BF$_4$], and (**g**–**i**) [BMIm][NTf$_2$] at 1:10, 1:30, and 1:50 exfoliated graphene hybrid oils and (**j**) the base oil.

### 3.5. Lubrication Mechanisms of Exfoliated Graphene

In order to explore the lubrication mechanism of exfoliated graphene products as additives, the worn surfaces lubricated with exfoliated graphene hybrid oils were characterized using Raman analysis, as shown in Figure 15. The Raman curves of all wear tracks show obvious D and G peaks. Meanwhile, the $I_D/I_G$ ratio was significantly higher than that of the samples before friction, indicating that the exfoliated graphene additives successfully entered the tribo-interface and participated in the processes of friction-reduction and wear resistance [20]. Related studies have shown that during the friction process, graphene modified with ILs usually forms a protective film at the friction interface and participates in friction reaction [39,40]. In order to verify this result, the additive debris of the $P_{10}$ hybrid oil with the best lubrication performance was collected, and the surface morphology and structure were observed under FESEM and a transmission electron microscope (TEM), as shown in Figure 16. In Figure 16a,b, the surface of the $P_{10}$ sample before friction is relatively smooth and has obvious graphite lamellar structure, while a large number of bumps and undulation appear on the wear debris surface, indicating that the structure underwent disordered transformation during friction [41]. The TEM images (Figure 16e,f) of wear debris illustrate the increase in the impurities on the surface of $P_{10}$ and the weakening of the electron diffraction ring. The above results show that the multilayer modified graphene exfoliated with ILs in an organic solution, when used as an additive, successfully participated in the friction reduction and anti-wear process and this was accompanied by disorder transformation of its structure.

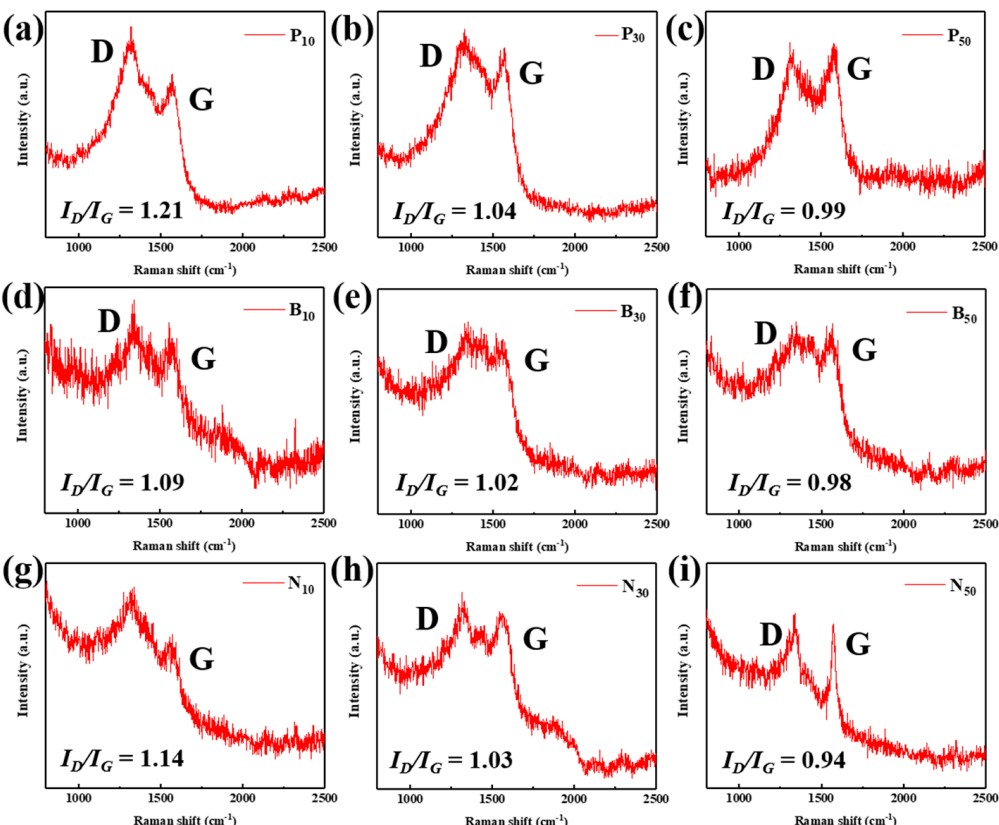

**Figure 15.** Raman curves of the worn surfaces lubricated with (**a–c**) [BMIm][PF$_6$], (**d–f**) [BMIm][BF$_4$], and (**g–i**) [BMIm][NTf$_2$] exfoliated graphene hybrid oils at 1:10, 1:30, and 1:50, respectively.

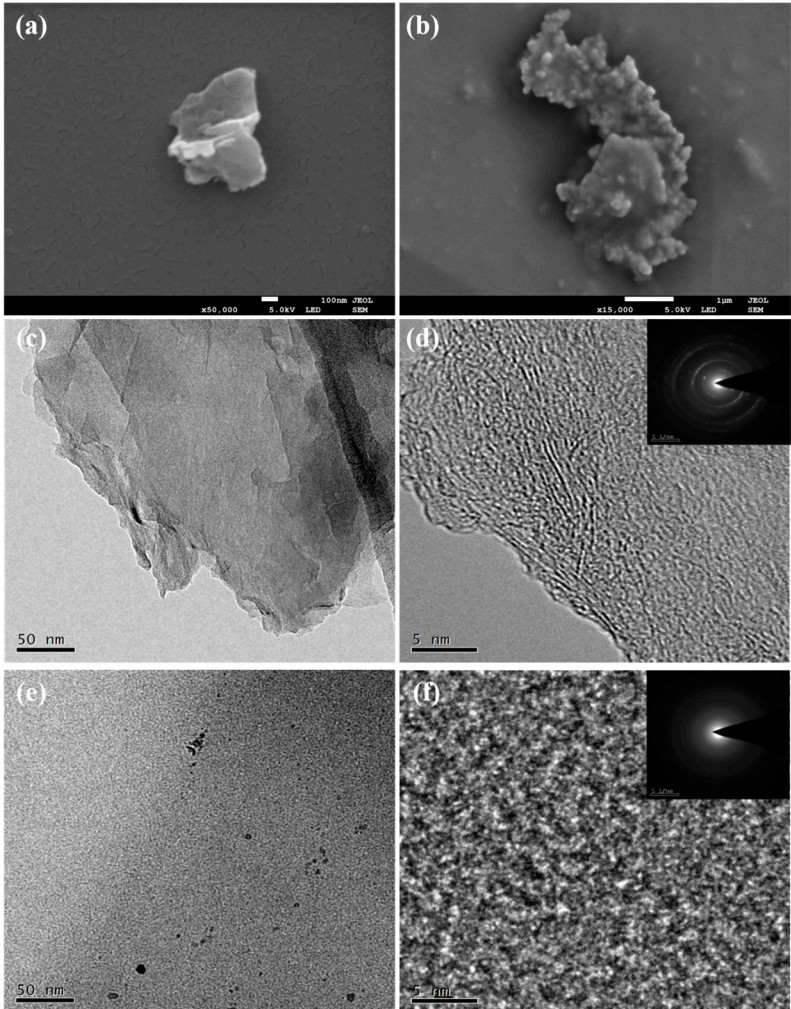

**Figure 16.** FESEM images of $P_{10}$ (**a**) before and (**b**) after friction; TEM, HRTEM, and SEAD images of $P_{10}$ (**c**,**d**) before and (**e**,**f**) after friction.

## 4. Conclusions

Electrochemically exfoliated graphene was successfully prepared using acetonitrile solutions of [BMIm][PF$_6$], [BMIm][BF$_4$], and [BMIm][NTf$_2$] ILs as electrolytes. Then the exfoliation mechanisms of the electrochemical modification were analyzed and explained. Moreover, the effects of the different IL at different concentrations on the morphology, chemistry, and tribological performance of the prepared exfoliated graphene were explored. Finally, the lubrication mechanisms of the exfoliated graphene as a lubrication additive were explored in detail. The conclusions can be drawn as follows:

(a) The modification density was proportional to the concentration of the ILs in the organic electrolyte. There were obvious differences in the modified graphene exfoliated in the different ILs in organic solvents, which may have been due to the different size of anions used.

(b) The hybrid oils containing modified graphene exfoliated with [BMIm][PF$_6$] and [BMIm][BF$_4$] ILs possessed high dispersion stability. The hybrid oil remained stable after 14 days.

(c) In contrast, the $P_{10}$ sample had the best antifriction and wear resistance, and the friction coefficient and wear rate were reduced by 32% and 39%, respectively.

(d) $P_{10}$ had a thinner layer, which could better enter the friction interface, to form a tribo-film, accompanied by the disorder transformation of its structure.

**Author Contributions:** Conceptualization, X.F. methodology, C.Z.; validation, X.F.; formal analysis, X.Z.; investigation, W.Z.; resources, X.F.; data curation, X.Z.; writing—original draft preparation, X.Z. and Z.Z.; writing—review and editing, X.F.; funding acquisition, X.F. All authors have read and agreed to the published version of the manuscript.

**Funding:** This research was funded by the National Natural Science Foundation of China, grant number No. 52075458 and U2141211.

**Institutional Review Board Statement:** Not applicable.

**Informed Consent Statement:** Not applicable.

**Data Availability Statement:** Not applicable.

**Acknowledgments:** The authors gratefully acknowledge the financial support provided by the Testing Center of Southwest Jiaotong University for providing testing services.

**Conflicts of Interest:** The authors declare no conflict of interest.

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
