# Peer review of "Functionalized Graphene from Electrochemical Exfoliation of Graphite toward Improving Lubrication Function of Base Oil"

_lubricants, doi:10.3390/lubricants11040166_

Round 1

Reviewer 1 Report

The manuscript is clear and simple to follow even though without the supplementary material available.

Please write the name related to the acronyms at the first time that you use it

Please provide the size of the steel ball

Figure 2 -please make the scale bar clear. In addition, the sheets that you present are quite thick to be considered graphene, for me they are small particles of graphite.

Please convert wear volume in wear coefficient (figure 12 e 13) and compare with values of other research works. In addition I suggest to compare your friction coefficient results with other research works.

Section 4 is not discussion but instead conclusions.

Author Response

Review 1: The manuscript is clear and simple to follow even though without the supplementary material available.

Question 1: Please write the name related to the acronyms at the first time that you use it

Answer 1: We really appreciate your careful review of the manuscript. We have carefully revised the manuscript and marked it in red.

Question 2: Please provide the size of the steel ball

Answer 2: Thank you very much. 

The diameter of GCr15 bearing steel as the upper steel ball (HRC 62) is 10 mm. The fixed lower steel plate (HRC 63, Ra = 28 nm) consists of AISI 52100 steel (Φ24 × 7.9 mm).

Question 3: Figure 2 -please make the scale bar clear. In addition, the sheets that you present are quite thick to be considered graphene, for me they are small particles of graphite.

Answer 3: Thanks a lot.

Thank you for your critical comments. We have made the scale bar clear. In the XRD analysis of graphene sheets, the (004) diffraction peak, which represents graphite, has disappeared. In addition, the 2D peak intensity of the sheet in the Raman spectrum increases significantly, which provides important information on whether the sheet is graphene or not. More importantly, TEM images of P10, B10 and N10 stripped graphene are shown in Figure 1. Obviously, it can be concluded that the sheets are multilayer graphene combined with the topography and electron diffraction.

Figure 1. TEM images and electron diffraction of P10, B10 and N10.

Question 4: Please convert wear volume in wear coefficient (figure 12 e 13) and compare with values of other research works. In addition, I suggest to compare your friction coefficient results with other research works.

Answer 4: Thanks a lot.

Thank you for your suggestions. The wear volume into wear rate have been converted. In addition, the tribological properties of graphene as a lubricant additive were compared with those in other literature, as shown in Table 1

Table 2. Tribological performance of lubricants reported by other literature, as comparison of this study.

Base oil

Lubricants

Friction reduction

Wear

reduction

Literature

Poly α-olefin

Engine oil

Hydraulic oil

PAO20

150N

graphene 

Ultrathin graphene

T-GO

E2C12-GO

graphene

5%

80%

50%

30%

32%

15%

33%

20%

57%

39%

[1]

[2]

[3]

[4]

This study

[1] Azman, S.S.N.; Zulkifli, N.W.M.; Masjuki, H.; Gulzar, M.; Zahid, R. Study of tribological properties of lubricating oil blend added with graphene nanoplatelets. J. Mater. Res. 2016, 31(13), 1932-1938.

https://doi.org/10.1557/jmr.2016.24. 

[2] Eswaraiah, V.; Sankaranarayanan, V.; Ramaprabhu, S. Graphene-based engine oil nanofluids for tribological applications. ACS Appl. Mater. Inter. 2011, 3(11), 4221-4227. https://doi.org/10.1021/am200851z

[3] Li, X.; Gan, C.; Han, Z.; Yan, H.; Chen, D.; Li, W.; Zhu, M. High dispersivity and excellent tribological performance of titanate coupling agent modified graphene oxide in hydraulic oil. Carbon 2020,165, 238-250.

https://doi.org/10.1016/j.carbon.2020.04.038

[4] Zhao, Z.; Fan, X.; Li, Y.; Zeng, Z.; Wei, X.; Lin, K.; Zhu, M. Well-dispersed graphene toward robust lubrication via reorganization of sliding interface. J. Ind. Eng. Chem. 2023, 119, 619-632.

https://doi.org/10.1016/j.jiec.2022.12.008

Question 5: Section 4 is not discussion but instead conclusions.

Answer 5: Thank you very much.

We apologize for this error in the manuscript. We have corrected our mistakes. Thank you again for your careful review.

Reviewer 2 Report

This research study the tribological performance of 150N base oil additivated with modified graphene exfoliated by three kinds of ILs with different concentrations. The article shows an interesting and clear good synthesis of modified graphene and very well characterized with different techniques such as tga, ftir, raman, xrd or xps. Furthermore, the tribological results are very interesting obtaining friction and wear reductions for all the studied nanolubricants in comparison to 150N base oil. The manuscript can be accepted after the following minor changes:

-Regarding the stability of the exfoliated graphene hybrid oils, it is very difficult to observe the sedimentation due to the dark color of the nanodispersions. It should be interesting to analyze the time stability through a quantitative method such as DLS or refractometry if it is possible.

-Improve the quality and presentation of figure 11.

Author Response

Review 2: This research study the tribological performance of 150N base oil additivated with modified graphene exfoliated by three kinds of ILs with different concentrations. The article shows an interesting and clear good synthesis of modified graphene and very well characterized with different techniques such as tga, ftir, raman, xrd or xps. Furthermore, the tribological results are very interesting obtaining friction and wear reductions for all the studied nanolubricants in comparison to 150N base oil. The manuscript can be accepted after the following minor changes:

Question 1: -Regarding the stability of the exfoliated graphene hybrid oils, it is very difficult to observe the sedimentation due to the dark color of the nanodispersions. It should be interesting to analyze the time stability through a quantitative method such as DLS or refractometry if it is possible.

Answer 1: Thanks a lot.

Zeta potential is a simple method to test the dispersion stability of nanoparticles. The increase of zeta potential indicates that the stability of nanofluids is better. The liquid layer of the fluid around the dispersed particles consists of two parts (Figure 1): one is the tail layer or the internal region with strong ion binding, and the other is the diffusion layer or the outer region with loose ion binding. There is an imaginary boundary in the diffusion layer in which ions or particles form a stable entity. When ions / particles cross the boundary under the action of gravity, they stay in the bulk phase, and the potential on the boundary surface is called zeta potential. The increase of electrostatic repulsion between nanoparticles will lead to the increase of zeta potential. However, the application of zeta potential to evaluate the stability of suspended nano-lubricants is limited. Zeta potential is not suitable for high viscosity base oils or dark lubricating oils.

Figure 1. Schematic diagram of electric double layer model.

When evaluating the dispersion stability of the graphene lubricant in the static test, the bottom precipitation of the bottle is the marker of the end of the static test. Based on the reviewer’s meaningful suggestions, we will evaluate the dispersion stability of graphene oil products by combining quantitative analysis, static test and ultraviolet absorption spectrum in the following studies.

Question 2: -Improve the quality and presentation of figure 11.

Answer 2: Thank you very much.

Thank you for your attention to detail, which makes the manuscript more rigorous. The quality and presentation of the figure 11 has been improved.

Thank you again for your excellent review and comments.
